# Chemical Characterization of Flowers and Leaf Extracts Obtained from *Turnera subulata* and Their Immunomodulatory Effect on LPS-Activated RAW 264.7 Macrophages

**DOI:** 10.3390/molecules27031084

**Published:** 2022-02-06

**Authors:** Jefferson Romáryo Duarte da Luz, Eder A. Barbosa, Thayse Evellyn Silva do Nascimento, Adriana Augusto de Rezende, Marcela Abbott Galvão Ururahy, Adriana da Silva Brito, Gabriel Araujo-Silva, Jorge A. López, Maria das Graças Almeida

**Affiliations:** 1Post-Graduation Program in Health Sciences, Health Sciences Center, Federal University of Rio Grande do Norte, R. Gen. Gustavo Cordeiro de Farias, s/n—Petrópolis, Natal 59012-570, RN, Brazil; jefferson_romaryo@hotmail.com (J.R.D.d.L.); adrirezende@yahoo.com (A.A.d.R.); 2Multidisciplinary Research Laboratory, DACT, Health Sciences Center, Federal University of Rio Grande do Norte, R. Gen. Gustavo Cordeiro de Farias, s/n—Petrópolis, Natal 59012-570, RN, Brazil; t_hayse_13@hotmail.com (T.E.S.d.N.); jorgejal@gmail.com (J.A.L.); 3Laboratory of Synthesis and Analysis of Biomolecules (LSAB), Institute of Chemistry, Darcy Ribeiro University Campus, University of Brasilia, Brasília 70910-900, DF, Brazil; bioederr@gmail.com; 4Post-Graduation Program in Pharmaceutical Sciences, Health Sciences Center, Federal University of Rio Grande do Norte, R. Gen. Gustavo Cordeiro de Farias, s/n—Petrópolis, Natal 59012-570, RN, Brazil; marcelaururahy@yahoo.com.br; 5Faculty of Health Sciences of Trairi (FACISA/UFRN), R. Passos de Miranda, Santa Cruz 59200-000, RN, Brazil; britoas.ufrn@gmail.com; 6Organic Chemistry and Biochemistry Laboratory, Amapá State University (UEAP), Av. Presidente Vargas, s/n, Centro, Macapá 68900-070, AP, Brazil; gabriel_ar4@yahoo.com.br

**Keywords:** plant, natural compounds, chromatography, anti-inflammatory

## Abstract

The anti-inflammatory properties of *Turnera subulata* have been evaluated as an alternative drug approach to treating several inflammatory processes. Accordingly, in this study, aqueous and hydroalcoholic extracts of *T. subulata* flowers and leaves were analyzed regarding their phytocomposition by ultrafast liquid chromatography coupled to mass spectrometry, and their anti-inflammatory properties were assessed by an in vitro inflammation model, using LPS-stimulated RAW-264.7 macrophages. The phytochemical profile indicated vitexin-2-*O*-rhamnoside as an important constituent in both extracts, while methoxyisoflavones, some bulky amino acids (e.g., tryptophan, tyrosine, phenylalanine), pheophorbides, and octadecatrienoic, stearidonic, and ferulic acids were detected in hydroalcoholic extracts. The extracts displayed the ability to modulate the in vitro inflammatory response by altering the secretion of proinflammatory (TNF-α, IL-1β, and IL-6) and anti-inflammatory (IL-10) cytokines and inhibiting the PGE-2 and NO production. Overall, for the first time, putative compounds from *T. subulata* flowers and leaves were characterized, which can modulate the inflammatory process. Therefore, the data highlight this plant as an option to obtain extracts for phytotherapic formulations to treat and/or prevent chronic diseases.

## 1. Introduction

Inflammation is a human body defense response to aggressive stimuli, involving biochemical, physiological, and immunological reactions to locate, inactivate, and destroy the offending agent, in addition to promoting tissue healing and repair [1,2]. During the inflammatory process, several cellular pathways play a crucial role in recovering and maintaining homeostatic balance. In this process, macrophage participation in the immune system is critical to the activation of inflammatory pathways and in the specific inflammatory mediator release [3].

Macrophages present antigens to cells, acting as central mediators in the immune system control, contributing to both the initiation and the resolution of inflammation. Activated macrophages secrete several inflammatory mediators, including cytokines such as TNF-α, IL-1β, IL-6, and IL-10, as well as PGE-2 and nitric oxide. Cytokines, such as TNF-α, IL-1β, and IL-6, activate the inflammatory response, while anti-inflammatory cytokines such as IL-10 act in the repair processes. Dysregulation of inflammatory mediator signaling interrupts this coordinated process and can lead to pathology. Accordingly, in inflammatory diseases, the cytokine network balance is severely disrupted, leading to persistent inflammation, tissue damage, and eventually organ failure [4,5].

Nonsteroidal anti-inflammatory drugs (NSAIDs) are used daily and indiscriminately in worldwide inflammatory disease treatment [6]. Overall, NSAIDs can cause side-effects due to their mechanism of action. Major adverse effects include gastrointestinal bleeding, changes in cardiovascular and renal functions, and other complications such as hemorrhage, perforation, or death [7]. Despite the diversity of anti-inflammatory drugs on the pharmaceutical market, no formulation offers low toxicity and minimal adverse effects. This situation stimulates the search for new molecules aiming at safer drugs with low side-effects for the inflammation treatment [8,9]. The mechanisms involved in the inflammatory response are fundamental for anti-inflammatory drug development [10,11].

In this context, plant phenolic compounds (e.g., flavonoids, phenolic acids, lignins) have been described regarding their several pharmacological activities, including antioxidant, anti-inflammatory, hepatoprotective, and antimicrobial effects [12]. Polyphenols are the object of studies to develop new drugs due to their direct or indirect inhibition or activation of important cellular and molecular targets, by modulating the expression of inflammatory mediators [13,14].

On this basis, studies have focused on evaluating plant biodiversity as a compound reservoir for therapeutic purposes, in addition to elucidating medicinal activities attributed by popular tradition [12,15]. In this context, *Turnera subulata* (Passifloraceae family) is a species with wide distribution in tropical and subtropical regions [16,17], used in folk medicine due to its pharmacological properties, such as anti-inflammatory, hypoglycemic, antifungal, and antioxidant activity [16,18,19,20,21]. Phytochemical studies with species of the *Turnera* genus have revealed the presence of flavanols, alkaloids, tannins, cyanogenic glycosides, fatty acids, triterpenoids, and various phenolic compounds related to their bioactivities [16,22].

Several studies have reported the use of plants from the Passifloraceae family for inflammatory process treatment [23,24]. Regarding *T. subulata*, their anti-inflammatory effects have mainly been described by inhibiting the cytokine production [25]. In a previous study, Luz et al. [26] showed the ability of *T. subulata* extracts to directly and indirectly inhibit thrombin, promoting few side-effects. These data are relevant since this coagulation factor is also related to an inflammatory response [27].

Hence, studies on the immunomodulatory potential of this plant species are relevant, in addition to proposing an association between the phytochemical profile and their pharmacological effects. Furthermore, this study is the first report on the phytochemical characterization and immunomodulatory effect of *T. subulata* flower extracts. Thus, this work chemically analyzed *T. subulata* floral and leaf extracts and evaluated their immunomodulatory effects in an in vitro model using RAW 264.7 macrophages stimulated by LPS, aiming to contribute to the search for anti-inflammatory drugs with low side-effects.

## 2. Results

The *T. subulata* hydroethanolic extracts displayed the highest yields (5.43% and 6.93% for flowers and leaves, respectively) compared to values of 4.16% and 5.3% obtained with flower and leaf aqueous extracts, respectively. This result was similar to those described by Antonio and Brito [18] for extracts from *T. ulmifolia* aerial parts, indicating the importance of the solvent choice for the extract yield.

*T. subulata* flower and leaf extracts were analyzed by UPLC–MS/MS, and their MS/MS spectra were submitted to compound identification using the GNPS database. Although a large number of MS/MS spectra were acquired for each extract that matched to known molecules deposited in the GNPS database, only those spectra with cosine ≥0.85 and mass difference ≤0.005 Da were considered. The extract phytochemical profile indicated the presence of vitexin-2-*O*-rhamnoside or its isoforms, a glycosylated flavone derivate from apigenin, as a constituent with potential pharmacological interest in all analyzed extracts. Furthermore, other compounds were tentatively identified, such as methoxyisoflavones, pheophorbides, octadecatrienoic, stearidonic, ferulic acids, and some bulky amino acids (e.g., tryptophan, tyrosine, phenylalanine) or their isoforms (Figure 1).

The extracted ion chromatograms (XICs) obtained for each of the of the structures tentatively identified by UPLC–MS/MS and GNPS analysis showed four major phytocomponents with good resolution for AFETS, (Figure 1A) five for ALETS, (Figure 1B) eight for HEFTS (Figure 1C), and eight for HELTS (Figure 1D). In all extracts, vitexin was the highest-intensity phytocomponent among the identified compounds; however, its exact sum still requires further quantification. Table 1 shows phytocomponents and their respective cosine, mass difference, and mass. The comparison between library GNPS and query spectra of the phytocomponents, as well as the draw structures identified, is presented in the Appendix A.

After determination of the chemical composition of AFETS, ALETS, HEFTS, and HELTS, their potential cytotoxic effects on RAW 264.7 macrophage cells were evaluated (Figure 2). The results showed no cytotoxic effects evaluated through cell viability assay by MTT (Figure 2A) and by Alamar Blue^®^ (Figure 2B), after exposing murine macrophages to different concentrations of these extracts.

It is known that macrophages are activated in the presence of LPS, which promotes the secretion of a high content of proinflammatory molecules, such as cytokines. The ELISA results indicated that the treatment with AFETS, ALETS, HEFTS, and HELTS significantly reduced the secretion of inflammatory cytokines (TNF-α and IL-1β) in LPS-activated murine RAW 264.7 macrophages in a concentration-dependent manner (Figure 3). AFETS and ALETS efficiently reduced the TNF-α secretion at 100 and 500 µg/mL, reaching 50% and 65% inhibition, respectively, while HEFTS and HELTS displayed a stronger action on this cytokine secretion at the same concentrations, reaching up to 75% inhibition (Figure 3A–D). Regarding IL-1β, the aqueous extracts inhibited its secretion by approximately 50%, although the hydroethanolic extracts at 500 µg/mL exhibited the highest percentage of inhibition (70%) (Figure 4A–D).

Additionally, all extracts were less effective in modulating IL-6 secretion, showing no statistically significant difference for AFETS and ALETS, whereas hydroethanolic extracts (HEFTS and HELTS) stabilized their inhibitory effects by 50% at 500 µg/mL (Figure 5A–D). Overall, both aqueous and hydroethanolic extracts of *T. subulata* significantly increased the anti-inflammatory cytokine IL-10 levels (Figure 6A–D). AFETS and ALETS displayed an increase of around 15% in IL-10 secretion, while HEFTS and HELTS achieved a 20% increase.

AFETS, ALETS, HEFTS, and HELTS cell treatments were also effective in inhibiting the prostaglandin E2 production (Figure 7A–D). AFETS promoted an inhibitory effect on prostaglandin E2 secretion at the two highest concentrations tested (100 and 500 µg/mL), reaching values around 50% and 60%, respectively. The inhibition induced by ALETS at 500 µg/mL was 50%. Additionally, AFETS and HEFTS displayed an inhibitory effect, stabilizing their action at 60% at the highest concentrations, while HELTS decreased the PGE-2 secretion by around 80% at 500 µg/mL.

Furthermore, the nitric oxide production was quantified to verify the anti-inflammatory activity of *T. subulata* extracts. The results indicated that both aqueous and hydroethanolic extracts at 100 and 500 µg/mL promoted a significant decrease (*p* < 0.05) in NO production by macrophages compared to the LPS-stimulated control group (Figure 8A–D). AFETS inhibited NO secretion by around 30% at the two highest concentrations tested (100 and 500 µg/mL), while ALETS only reached this inhibitory percentage at 500 µg/mL. HEFTS decreased NO secretion by 30% and 40% at concentrations of 100 and 500 µg/mL, respectively. No statistically significant difference was observed compared to the negative control. Regarding HELTS, the results showed that the highest concentrations tested (100 and 500 µg/mL) inhibited nitric oxide secretion by 40% and 65%, respectively, again with no statistical difference compared to the negative control.

## 3. Discussion

Studies on the regulatory mechanisms of inflammatory responses have revealed that inflammation can delay healing and, in a chronic situation, can also play an important role in the onset of cancer and other chronic diseases [28]. Therefore, the use of nonsteroidal anti-inflammatory drugs for inflammation suppression is prevalent, despite several adverse effects, such as gastrointestinal bleeding and ulcers [29]. On the other hand, increasing evidence suggests that natural products, such as plant extracts, due to their phytocomposition, can decrease inflammatory processes [14,30,31].

The properties of several secondary plant metabolites have been extensively studied, mainly due to their antioxidant and anti-inflammatory properties [32,33,34]. Thus, this study presented the chemical profile characterization of aqueous and hydroethanolic extracts of *T. subulata* flower and leaf by mass spectrometry, as well as the evaluation of their anti-inflammatory properties in an in vitro model. At this point, it is noteworthy that only a few studies have reported the presence of phenolic compounds especially in *T. subulata* flowers, as well as their pharmacological properties [25,26,35].

Regarding the phytocomposition, it is noteworthy that phenolic compounds are the main class among the secondary metabolites present in *Turnera* genus [22]. Accordingly, *T. subulata* flower and leaf extract analyses revealed the presence of three phenolic compounds: vitexin-2-*O*-ramnhoside as a valuable chemical constituent, as well as 7-*O*-beta-glucopyranosyl-4′-hydroxy-5-methoxyisoflavone and ferulate. Although quantitative experiments need to be performed to validate the potential relevance of vitexin-2-*O*-ramnhoside in the *T. subulata* flower and leaf extracts, this phytochemical has already been characterized as the main constituent in extracts of distinct plants from the Magnoliopsida class [36,37,38,39,40]. These phenolic compounds are scarce in the genus *Turnera*, and this is the first report on the identification of these secondary metabolites for the genus. Studies have only identified the presence of other secondary metabolites in *Turnera* leaves, flowers, and roots [22], such as four different lutein structures and two apigenin structures [41], as well as naringenin, three apigenin coumaroyl glucosides, and five flavone aglycones [42]. The presence of these flavonoids in *T. subulata* extracts is possibly related to the climatic conditions of the Brazilian caatinga biome, a place of flower and leaf collection, where high exposure to solar radiation favors the biosynthesis of phenolic compounds such as flavones [43].

Anti-inflammatory properties observed with treatments using *T. subulata* extracts are correlated with the flavonoid presence, which points to the involvement of these compounds in inhibiting the synthesis and activity of different proinflammatory mediators [44]. Furthermore, flavonoids are reported as molecules with low cytotoxicity [44,45]. Therefore, the presence of these phytocomponents in the AFETS, ALETS, HEFTS, and HELTS could be related to the anti-inflammatory effect observed in this study. These *T. subulata* extracts exhibited an in vitro effect, evaluated by the inhibition of proinflammatory cytokines (TNF-α, IL-1β, and IL-6) in LPS-stimulated macrophages. Moreover, these extracts stimulated anti-inflammatory cytokine IL-10 secretion under the same experimental conditions.

These immunomodulatory effects may be associated with metabolites tentatively identified in the extracts, since vitexin is known to have an anti-inflammatory action by inhibiting important cytokines involved in the inflammatory cascade [46]. In a colitis experimental model of colitis, vitexin significantly inhibited the TNF-α, IL-6, and IL-1β expression, suggesting that this compound may suppress the inflammatory response to improve colitis-induced liver damage [47]. Another study showed that vitexin decreased the MCP-1 (a monocytic cytokine), IL-6, and IL-8 levels, while concomitantly increasing IL-10 expression in a murine model of septic encephalopathy [48].

Isoflavones are other metabolites with anti-inflammatory effects described in the literature. Thus, milletenol A, an isoflavone isolated from *Milletia pacchycarpa* seeds, decreased the neutrophil numbers in an inflammation model of zebrafish stimulated by copper sulfate [49]. Furthermore, isoflavones identified from *Glycine max* leaves exhibited positive effects by reducing the production of IL-1β and IL-6 in LPS-stimulated RAW 264.7 cells [50].

Regarding ferulic acid also detected in *T. subulata* extracts, studies have demonstrated its anti-inflammatory action by suppressing the expression of proinflammatory cytokines (TNF-a, IL-1B, IL-6, and IL-8) in LPS-induced primary cells isolated from bovine uterus via inhibition of the NF-κB and MAPK pathways as the main mechanism of action. Additionally, ferulic acid reduced TNF-a, IL-1b, and NF-κB secretion in an in vivo model of liver injury induced by methotrexate in rats [51,52].

During the proinflammatory response in LPS-stimulated cells, cytokines such as IL-1β, IL-6, IL-10, and TNF-α are released. The expression and release of these cytokines can be regulated by different transcription factors, such as NF-κB, and they are usually associated with metabolic processes [53]. Blockade of IL-1β action/secretion has been associated with anti-inflammatory responses [54], which may suggest that *T. subulata* extracts may also act on these metabolic pathways, since the extracts modulated the secretion of the same cytokines.

According to the chemical composition, other mechanisms of action may be associated with the immunomodulatory effects observed with aqueous and hydroethanolic extracts of *T. subulata* flowers and leaves. One mechanism suggested that the anti-inflammatory effect of this species may be due to the ability of leaf extracts to modulate MAPK signaling pathways by inhibiting ERK1/2 phosphorylation and blocking the inflammatory response in RAW 264.7 macrophages. This effect also decreases TNF-α and IL-1β secretion, increasing the blockade of RAGE and CD40 as the main mechanisms of action [25].

Inflammatory mediators, such as PGE-2 and NO, and several proinflammatory cytokines released by activated macrophages are important targets for inflammatory disease treatment, including multiple sclerosis and rheumatoid arthritis. When overexpressed, these proinflammatory mediators can lead to cell and tissue damage, resulting in various physiological disorders related to inflammation [55,56]. Therefore, compounds that inhibit the production of proinflammatory mediators such as PGE-2 and NO can be candidates as anti-inflammatory drugs [57].

The AFETS, ALETS, HEFTS, and HELTS extracts reduced the PGE-2 and nitric oxide levels by more than 60%. This study shows for the first time that *T. subulata* extracts as active natural products exhibit inhibitory effects on the secretion of these proinflammatory mediators, which may also be associated with the phenolic groups detected in the extract chemical composition. According to studies, the polyphenol combination displays synergistic effects on biological activity [58,59,60].

Regarding its ability to modulate cytokine secretion, the presence of vitexin may also play an important role in the *T. subulata* extract in reducing PGE-2 and NO levels. This is consistent with studies showing vitexin’s capacity to decrease the PGE-2 levels in cultured chondrocytes from osteoarthritic patients [46], as well as NO production associated with the autophagic dysfunction of ischemic stroke, and to protect against cerebral endothelial permeability [61,62]. An analogous situation can be verified with isoflavones also identified in *T. subulata* extracts and their association with the modulation of NO, compared to the concentration-dependent effect exhibited by the isoflavone milletenol A, which was able to inhibit NO secretion in an in vitro assay using LPS-stimulated RAW 264.7 macrophages [50]. In the inflammatory process, the increase in PGE-2 levels is stimulated by the NO production, which is closely associated with an increase in TNF-α production [63,64]. In the present study, the TNF-α secretion was inhibited by all *T. subulata* extracts, with a correlated decrease in NO levels and, consequently, PGE-2 levels, suggesting a well-controlled mechanism of action.

In addition to the immunomodulatory effects demonstrated in this study, it is noteworthy that Luz et al. [26] showed the ability of *T. subulata* extracts to inhibit thrombin, the main protease in the coagulation cascade, which plays a role in the inflammatory response [65]. Thrombin can signal the expression of proinflammatory cytokines, such as IL-1β and IL-6, and cell adhesion molecules, promoting leukocyte activation and fibroblast proliferation, as well as contributing to leukocyte recruitment [65,66]. The studies with *T. subulata* ethanol and ethyl acetate leaf extracts demonstrated a direct inhibitory effect on thrombin and an indirect one through heparin cofactor II, corroborating the effect of the extracts on the inflammatory process [26].

Overall, these results support ethnopharmacological studies, which describe the use of flowers and leaves of the genus *Turnera* in folk medicine to treat inflammatory diseases [22]. Experimental data indicate the immunomodulatory effect of aqueous and hydroalcoholic extracts of *T. subulata* flowers and leaves through the inhibition of inflammatory cytokine secretion and an increase in anti-inflammatory cytokine IL-10 levels. Despite the scarce data in the literature regarding extracts of the species *T. subulata*, it is possible to suggest its anti-inflammatory action. Accordingly, this is a pioneering study showing the chemical characterization of its flower and leaf extracts associated with potential therapeutic effects.

## 4. Material and Methods

### 4.1. Plant Material and Extract Preparation

*T. subulata* flowers and leaves were collected in Natal, Rio Grande do Norte, Brazil. The species were taxonomically identified by Dr. Jomar Gomes Jardim. A voucher specimen (No. 0674/08) was deposited in the Herbarium at the Department of Botany and Zoology, Federal University of Rio Grande do Norte, Natal, RN, Brazil. The aerial parts of *T. subulata* were air-dried at 40 °C for 48 h and powdered to particle size <180 µm. Then, 300 g of powdered flowers and leaves were individually subjected to decoction in water (100 °C/10 min), filtered on Whatman filter papers, and lyophilized to obtain the aqueous flower extract (AFETS) and aqueous leaf extract (ALETS). Regarding the hydroethanolic extracts, 300 g of flowers and leaves were individually extracted by maceration with 1.5 L of ethanol/water (50:50, *v*/*v*) for 4 days at room temperature. Extracts were filtered, rotary evaporated, and lyophilized; they were denominated as hydroethanolic flower extract (HEFTS) and hydroethanolic leaf extract (HELTS).

### 4.2. Chemical Characterization by Ultrafast Liquid Chromatography Coupled to Mass Spectrometry (LC–MS/MS)

All extract samples were initially reconstituted in methanol (µg/mL), centrifuged (30 min/13,000 rpm), and filtered through a 0.22 mm membrane. Their respective supernatants were stored at −20 °C and then diluted in mobile phase (pure acetonitrile) to 10× and 20× for aqueous and ethanolic extracts, respectively, prior to LC–MS/MS analyses.

Extracts were analyzed by ultrafast liquid chromatography in a UPLC Eksigent UltraLC 110-XL liquid chromatograph (AB Sciex, Framingham, MA, USA) coupled to Kinetex 2.6 μm C18 100 Å column (50 × 2.1 mm) and a 5600+ TripleT spectrometer (AB Sciex, Framingham, MA, USA). After equilibrating the column with 5% acetonitrile/0.1% formic acid solution for 5 min, 2 µL of the sample was automatically injected, and the separation was performed with a linear gradient of 5% acetonitrile/0.1% formic acid ranging 5–95% over 10 min at a flow rate of 0.4 mL/min, keeping the column temperature at 40 °C. The mass spectrometer operated in positive IDA (information dependent acquisition) mode with mass range from *m*/*z* 100 to 1800 and source temperature of 650 °C. The IDA mode was configured to fragment ions from *m*/*z* 100–1250, with a load ranging from 1 to 3, and with an intensity greater than 1000 counts. The other acquisition parameters were as follows: period cycle time = 900 ms; pulser frequency = 15,392 kHz; accumulation time = 250.0 ms; curtain gas = 15,000; ion source gas 1 = 50,000; ion source gas 2 = 45,000; ion spray voltage floating = 5500. In addition to the extracts, a blank control was acquired. Before the start and every five analyses, the spectrometer was calibrated using the calibration solution (AB Sciex, Framingham, MA, USA) to obtain an accuracy of approximately 0.5 ppm (sodium iodide (2 μg/μL) and cesium iodide (50 ng/μL) in 50/50 2-propanol/water).

For data analysis, acquisition files (.WIFF) were converted to the .mzXML format using the MSConvert software (ProteoWizard 3.0, ProteoWizard, Palo Alto, CA, USA) and submitted to the GNPS platform: Global Natural Products Social Molecular Networking (http://gnps.ucsd.edu) for analysis with the Molecular-Library Search-V2 (version release_14) tool (1). Data were filtered by removing all peaks with ~17 Da referring to the *m*/*z* value of the precursor present in the MS/MS spectra, which were filtered by window choosing only the top six peaks in the 50 Da window across the spectrum. Data were then grouped with the MS-Cluster with an original mass tolerance of 0.02 Da and an ion tolerance of MS/MS fragments of 0.1 Da to create consensus spectra. Furthermore, consensus spectra containing fewer than two spectra were discarded, before analysis in the GNPS spectral libraries. The library spectra were filtered in the same way as the input data. All correspondences maintained between network and library spectra were required to have a score above 0.85 and at least four corresponding peaks. Cosine score refers to a normalized dot-product, a mathematical measure of spectral similarity between two fragmentation spectra. A cosine score of 1 represents identical spectra, whereas a cosine score of 0 denotes no similarity at all [67].

### 4.3. Cell Culture

Murine macrophage (RAW 264.7) cells were obtained from American Type Culture Collection (ATCC, Rockville, MD, USA) (ATCC^®^ TIB-71™). Cells were grown in DMEM (Dulbecco’s modified Eagle’s medium), supplemented with 10% fetal bovine serum, and streptomycin (5000 mg/mL)/penicillin (5000 IU) at 37 °C in a humidified atmosphere with 5% CO_2_.

### 4.4. MTT Viability Assay

RAW 264.7 macrophages were exposed to AFETS, ALETS, HEFTS, and HELTS at different extract concentrations (5, 50, 100, and 500 µg/mL) in order to assess the cytotoxicity. Cell viability was determined using the MTT assay (3-(4,5-dimethylthiazol-2-yl)-2,5-diphenyltetrazolium bromide). A total of 1 × 10^5^ cells per well were seeded in 96-well microplates for 24 h to promote adhesion. After 24 h at 37 °C, 100 μL of MTT (5 mg/mL) was added to each well containing cells, and the plates were again incubated (37 °C/4 h). After removing the culture medium, 100 µL of DMSO was added to each well, before determining the cell viability at 570 nm in a microplate ELISA reader (Epoch-Biotek, Winooski, VT, USA). Cells grown only in DMEM medium were used as a negative control. All assays were performed in triplicate.

### 4.5. Alamar Blue^®^ Viability Assay

Cell viability after challenge with AFETS, ALETS, HEFTS, and HELTS at 5, 50, 100, and 500 µg/mL was also evaluated by Alamar Blue^®^ assay to assess their cytotoxic effect. A total of 1 × 10^5^ cells per well were seeded in 96-well microplates for 24 h to promote adhesion. After 24 h at 37 °C of exposure to the extracts, 10% Alamar Blue^®^, corresponding to the volume of the medium contained in each well, was added, and the plate was again incubated (4 h/37 °C/5% CO_2_). Then, the reduced Alamar Blue^®^ was monitored at 570 nm and 600 nm in a microplate ELISA reader (Epoch-Biotek, Winooski, VT, USA). Cells grown only in DMEM medium were used as a negative control. All assays were performed in triplicate.

### 4.6. Cytokine Measurement of (TNF-α, IL1-β, IL-6, and IL10)

Raw 264.7 cells (1 × 10^5^ cells per well) were plated and activated with lipopolysaccharide (LPS) from *Escherichia coli* (O55:B5), previously dissolved in DMEM (2 µg/mL). After 1 h, cells were treated with AFETS, ALETS, HEFTS, and HELTS at different concentrations (5, 50, 100, and 500 µg/mL). After 24 h, supernatants were harvested, and the levels of TNF- α, IL1-β, IL-6, and IL-10 were measured using an immunoenzymatic assay (ELISA) kit (eBioscience), according to the manufacturer’s instructions. Analyses were performed in triplicate, and optical density was measured at 450 nm in a microplate ELISA reader (Epoch-Biotek, Winooski, VT, USA). Cells without LPS stimulation and LPS-stimulated cells without extract exposure were used as negative and positive control, respectively.

### 4.7. Measurement of Prostaglandin E_2_ (PGE_2_) and Nitric Oxide (NO) Production

Raw 264.7 cells (1 × 10^5^ cells per well) were plated and activated with LPS from *Escherichia coli* (O55:B5), dissolved in DMEM (2 µg/mL). One hour later, cells were treated with different concentrations (5, 50, 100, and 500 µg/mL) of AFETS, ALETS, HEFTS, and HELTS. After 24 h, supernatants were collected to determine PGE_2_ and NO, using ELISA kits according to the respective manufacturer’s instructions (BD Biosciences, San Jose, CA, USA). The PGE_2_ concentrations, expressed in pg/mL, were monitored at 405 nm and 420 nm, while the NO total concentration was assessed after addition of Griess reagent to 40 µL of supernatant and measuring the absorbance at 545 nm. All readings were performed in a microplate ELISA reader (Epoch-Biotek, Winooski, VT, USA). Cells without LPS stimulation and LPS-stimulated cells without extract exposure were used as the negative and positive control, respectively.

### 4.8. Statistical Analysis

The results were expressed as the mean ± SD and analyzed with one-way ANOVA and Tukey’s post hoc test, considering *p* < 0.05 as statistically significant. All statistical analyses were performed using GraphPad Prism version 5.0 Software for Windows (GraphPad Software, San Diego, CA, USA).

## 5. Conclusions

This study demonstrated that *T. subulata* and its bioactive molecules display promising anti-inflammatory activity and no cytotoxicity toward RAW 264.7 cells. The anti-inflammatory properties of *T. subulata* extracts were suggested using an in vitro model of acute inflammation through a reduction in TNF-α, IL-1β, and IL-6 cytokines and an increase in IL-10 cytokine. The extracts also inhibited PGE_2_ and NO production in LPS-stimulated RAW 264.7 cells. Phytochemical analyses revealed the presence of vitexin-2-*O*-rhamnoside or its isoforms, a glycosylated flavone derived from apigenin, which may be responsible for the pharmacological activities such as the immunomodulatory effect displayed by *T. subulata* extracts highlighted in this study. Overall, the results provide scientific evidence supporting the use of *T. subulata* flowers and leaves in folk medicine for inflammatory disorders, which can be applied as an alternative to assist with this treatment.

## Figures and Tables

**Figure 1 molecules-27-01084-f001:**
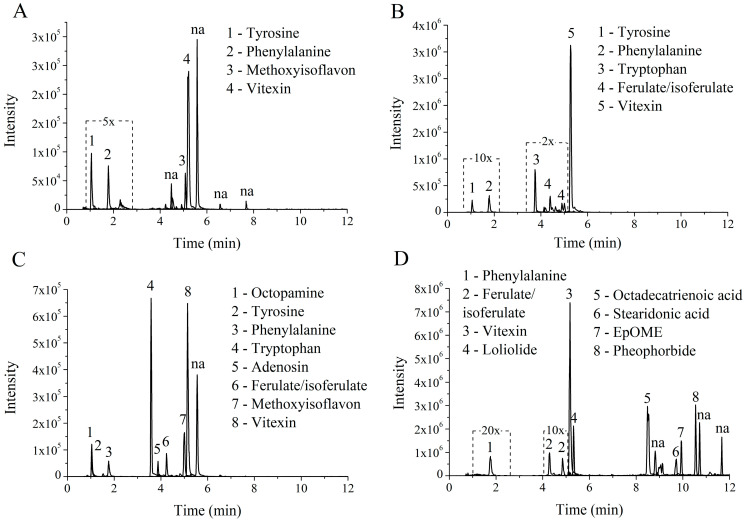
LC–MS/MS fingerprint of *Turnera subulata* extracts: (**A**) *T. subulata* aqueous flower extract (AFETS); (**B**) *T. subulata* aqueous leaf extract (ALETS); (**C**) *T. subulata* hydroethanolic flower extract (HEFTS); (**D**) *T. subulata* hydroethanolic leaf extract (HELTS). 2×, 5×, and 10× denote the magnification applied in dotted areas of the chromatogram; na = not available.

**Figure 2 molecules-27-01084-f002:**
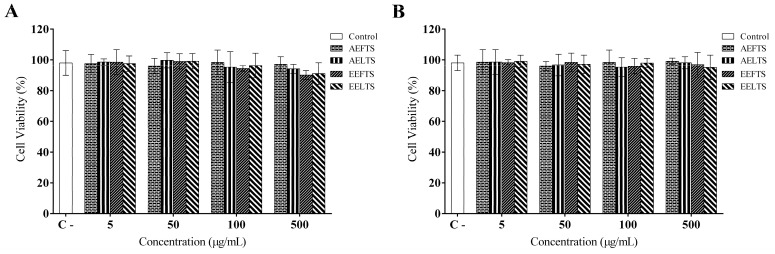
Cytotoxicity effects of AFETS, ALETS, HEFTS, and HELTS on RAW 264.7 murine macrophage cells: (**A**) cell viability measured by MTT assay; (**B**) cell viability measured by Alamar Blue assay. Culture medium DMEM was used as a negative control for cytotoxicity.

**Figure 3 molecules-27-01084-f003:**
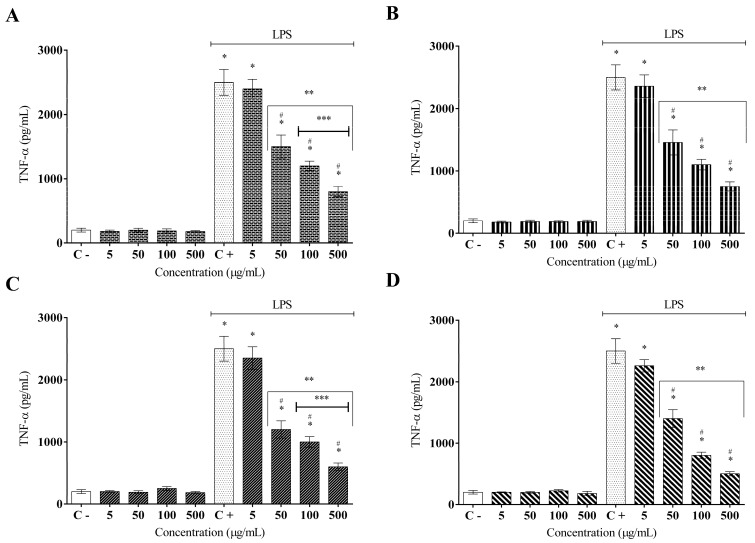
Effect of *Turnera subulata* extracts on the TNF-α cytokine release: (**A**) AFETS; (**B**) ALETS; (**C**) HEFTS; (**D**) HELTS. The cytokine content was released in RAW 264.7 cells and stimulated by LPS after 24 h. Release of cytokines was performed using ELISA assays. Data represent the mean ± SEM from three independent experiments. One-way ANOVA followed by the post hoc Tukey test. * *p* < 0.05 vs. the control group; # *p* < 0.05 vs. the LPS-stimulated cells; ** *p* < 0.05 between the concentrations of the extract; *** *p* < 0.05 between the higher concentrations (100 and 500 µg/mL).

**Figure 4 molecules-27-01084-f004:**
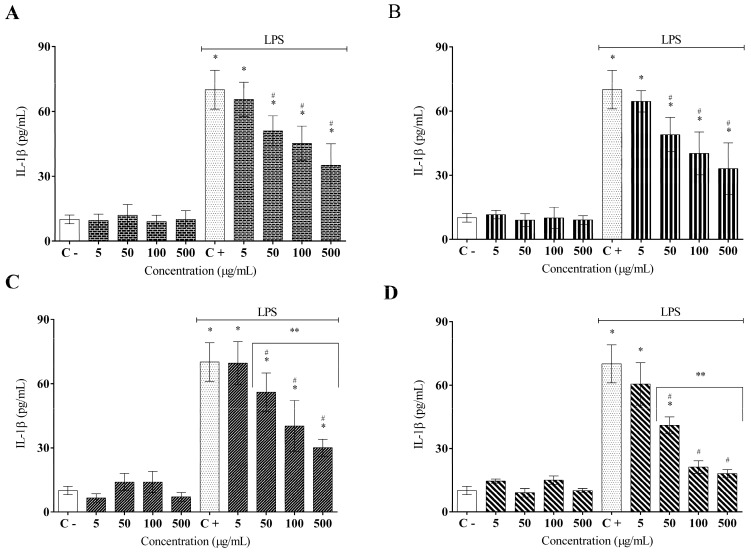
Effect of *Turnera subulata* extracts on the IL-1β cytokine release: (**A**) AFETS; (**B**) ALETS; (**C**) HEFTS; (**D**) HELTS. The cytokine content was released in RAW 264.7 cells and stimulated by LPS after 24 h. Release of cytokines was performed using ELISA assays. Data represent the mean ± SEM from three independent experiments. One-way ANOVA followed by the post hoc Tukey test. * *p* < 0.05 vs. the control group; # *p* < 0.05 vs. the LPS-stimulated cells; ** *p* < 0.05 between the concentrations of the extract.

**Figure 5 molecules-27-01084-f005:**
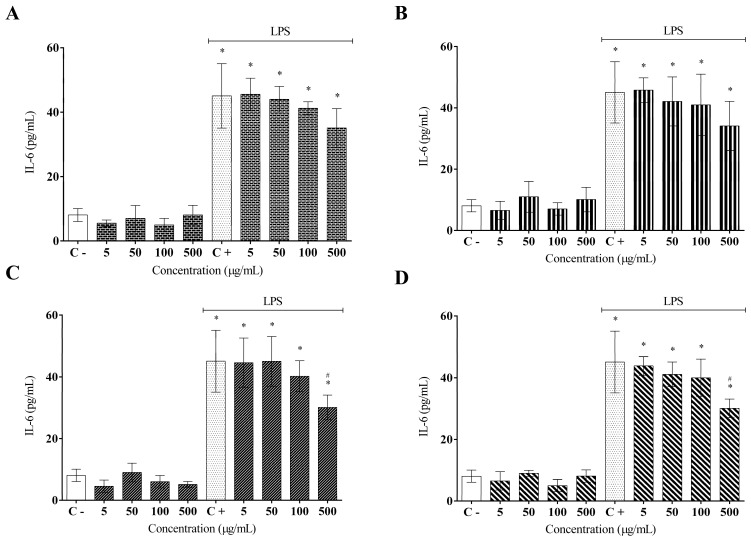
Effect of *Turnera subulata* extracts on the IL-6 cytokine release: (**A**) AFETS; (**B**) ALETS; (**C**) HEFTS; (**D**) HELTS. The cytokine content was released in RAW 264.7 cells and stimulated by LPS after 24 h. Release of cytokines was performed using ELISA assays. Data represent the mean ± SEM from three independent experiments. One-way ANOVA followed by the post hoc Tukey test. * *p* < 0.05 vs. the control group; # *p* < 0.05 vs. the LPS-stimulated cells.

**Figure 6 molecules-27-01084-f006:**
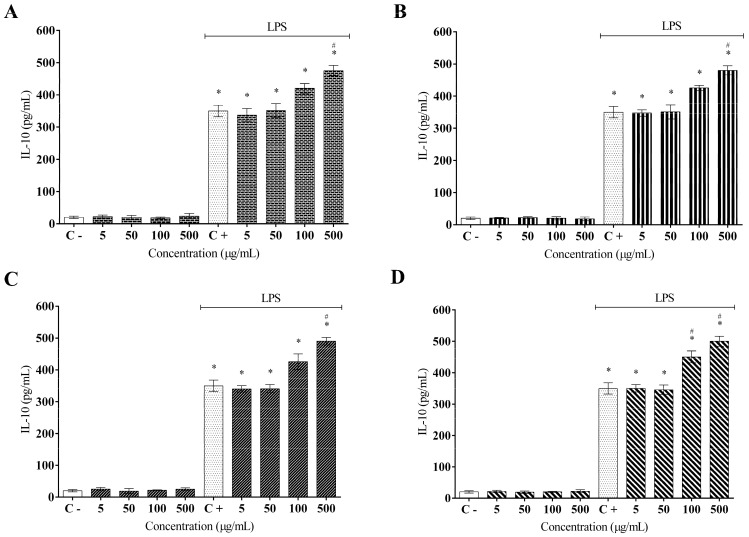
Effect of *Turnera subulata* extracts on the anti-inflammatory IL-10 cytokine release: (**A**) AFETS; (**B**) ALETS; (**C**) HEFTS; (**D**) HELTS. The cytokine content was released in RAW 264.7 cells and stimulated by LPS after 24 h. Release of cytokines was performed using ELISA assays. Data represent the mean ± SEM from three independent experiments. One-way ANOVA followed by the post hoc Tukey test. * *p* < 0.05 vs. the control group; # *p* < 0.05 vs. the LPS-stimulated cells.

**Figure 7 molecules-27-01084-f007:**
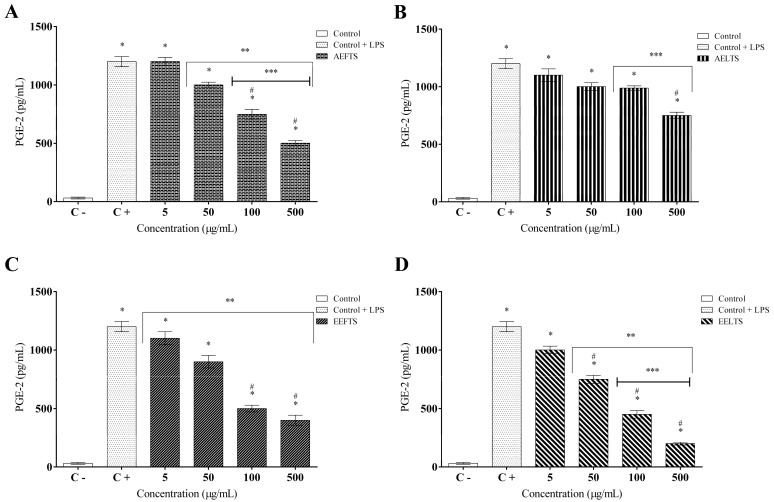
Inhibitory effects of *Turnera subulata* extracts on LPS-stimulated PGE-2 production in RAW 264.7 macrophages: (**A**) AFETS; (**B**) ALETS; (**C**) HEFTS; (**D**) HELTS. The level of PGE-2 in the culture medium was quantified using enzyme-linked immunoassay (ELISA) kits. Data represent the mean ± SEM from three independent experiments. One-way ANOVA followed by the post hoc Tukey test. * *p* < 0.05 vs. the control group; # *p* < 0.05 vs. the LPS-stimulated cells; ** *p* < 0.05 between the concentrations of the extract; *** *p* < 0.05 between the higher concentrations (100 and 500 µg/mL).

**Figure 8 molecules-27-01084-f008:**
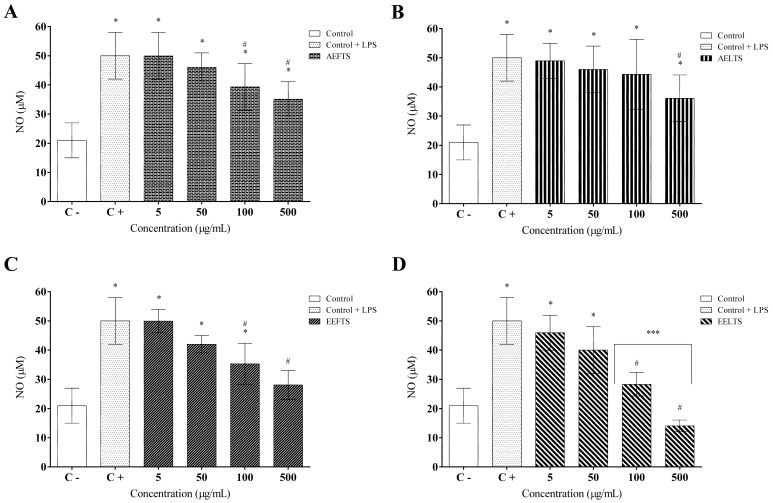
Inhibitory effects of *Turnera subulata* extracts on LPS-stimulated nitric oxide (NO) production in RAW 264.7 macrophages: (**A**) AFETS; (**B**) ALETS; (**C**) HEFTS; (**D**) HELTS. The level of NO in the culture medium was quantified using Griess reagent. Data represent the mean ± SEM from three independent experiments. One-way ANOVA followed by the post hoc Tukey test. * *p* < 0.05 vs. the control group; # *p* < 0.05 vs. the LPS-stimulated cells; *** *p* < 0.05 between the higher concentrations (100 and 500 µg/mL).

**Table 1 molecules-27-01084-t001:** Phytocomponents identified in *Turnera subulata* flower and leaf extracts by LC–MS/MS analyses.

Peak.	Phytocomponents Matched with GNPS Data Base	Cosine	Mass Diff	Mass	Molecular Formula	Ion Fragments ^a^	Adduct	Extract
1	l-Tyrosine	0.86	0.001	165.054	C_9_H_8_O_3_	147.04, 123.05, 119.05, 95.05, 91.06	[M − NH_3_ + H]^+^	AFETS
2	Phenylalanine	0.95	0	166.086	C_9_H_11_NO_2_	149.06, 131.05, 120.08, 103.05, 53.04	[M + H]^+^	AFETS
3	7-*O*-beta-glucopyranosyl-4′-hydroxy-5-methoxyisoflavone	0.89	0	447.129	C_22_H_22_O_10_	285.08, 270.05, 213.05, 152.01	[M + H]^+^	AFETS
4	Vitexin-2-*O*-rhamnoside	0.95	0.002	579.172	C_27_H_30_O_14_	433.11, 415.10, 397.09, 313.07, 283.06	[M + H]^+^	AFETS
1	l-Tyrosine	0.95	0.001	182.081	C_9_H_11_N_1_O_3_	165.06, 147.04, 136.07, 123.05, 119.05	[M + H]^+^	ALETS
2	dl-Phenylalanine	0.97	0	166.086	C_9_H_11_NO_2_	149.06, 131.05, 120.08, 103.05, 53.04	[M + H]^+^	ALETS
3	l-Tryptophan	0.95	0.001	205.096	C_11_H_12_N_2_O_2_	188.07, 159.09, 146.06, 144.08, 118.06	[M + H]^+^	ALETS
4	Ferulate/isoferulate	0.93	0	177.054	C_10_H_8_O_3_	149.06, 145.03, 117.03, 89.04	M − H_2_O + H	ALETS
5	Vitexin-2-*O*-rhamnoside	0.94	0	579.17	C_27_H_30_O_14_	433.11, 415.10, 397.09, 313.07, 283.06	[M + H]^+^	ALETS
1	dl-Octopamine	0.95	0	136.076	C_8_H_9_NO_1_	119.05, 118.06, 107.05, 91.06, 64.04	[M − H_2_O + H]^+^	HEFTS
2	l-Tyrosine	0.94	0.001	165.054	C_9_H_8_O_3_	147.04, 123.05, 119.05, 95.05, 91.06	[M − NH_3_ + H]^+^	HEFTS
3	Phenylalanine	0.96	0	166.086	C_9_H_11_NO_2_	149.06, 131.05, 120.08, 103.05, 53.04	[M + H]^+^	HEFTS
4	l-Tryptophan	0.94	0.001	188.07	C_11_H_9_N_1_O_2_	170.06, 146.06, 144.08, 143.07, 118.07	[M − NH_3_ + H]^+^	HEFTS
5	Adenosine, 5_-*S*-methyl-5_-thio-	0.91	0.004	298.097	C_11_H_15_N_5_O_3_S	145.03, 136.06, 97.03, 61.01	[M + H]^+^	HEFTS
6	Ferulate/isoferulate	0.89	0	177.054	C_10_H_8_O_3_	149.06, 145.03, 117.03, 89.04	[M − H_2_O + H]^+^	HEFTS
7	7-*O*-beta-glucopyranosyl-4′-hydroxy-5-methoxyisoflavone	0.91	0.001	447.13	C_22_H_22_O_10_	285.08, 270.05, 213.05, 152.01	[M + H]^+^	HEFTS
8	Vitexin-2-*O*-rhamnoside	0.97	0.003	579.173	C_27_H_30_O_14_	433.11, 415.10, 397.09, 313.07, 283.06	[M + H]^+^	HEFTS
1	Phenylalanine	0.94	0	166.086	C_9_H_11_NO_2_	149.06, 131.05, 120.08, 103.05, 53.04	[M + H]^+^	HELTS
2	Ferulate/isoferulate	0.94	0	177.054	C_10_H_8_O_3_	149.06, 145.03, 117.03, 89.04	[M − H_2_O + H]^+^	HELTS
3	Vitexin-2-*O*-rhamnoside	0.94	0.001	579.171	C_27_H_30_O_14_	433.11, 415.10, 397.09, 313.07, 283.06	[M + H]^+^	HELTS
4	Loliolide	0.94	0	197.117	C_11_H_16_O_3_	179.11, 161.09, 135.12, 133.10, 107.09	[M + H]^+^	HELTS
5	9*S*-Hydroxy-10*E*,12*Z*,15*Z*-octadecatrienoic acid	0.86	0	277.216	C_18_H_28_O_2_	259.20, 149.13, 135.12, 121.10, 93.07	[M − H_2_O + H]^+^	HELTS
6	9(10)-EpOME	0.91	0.002	279.233	C_18_H_30_O_2_	173.13, 109.10, 95.09, 81.07, 67.06	[M − H_2_O + H]^+^	HELTS
7	Stearidonic acid Ethyl ester	0.86	0.001	305.248	C_20_H_32_O_2_	259.20, 149.13, 135.12, 121.10, 93.07	[M + H]^+^	HELTS
8	Pheophorbide A	0.87	0.005	593.274	C_35_H_36_N_4_O_5_	533.25, 460.23, 447.22, 433.24, 431.18	[M + H]^+^	HELTS

*Turnera subulata* aqueous flower extract (AFETS); *T. subulata* aqueous leaf extract (ALETS); *T. subulata* hydroethanolic flower extract (HEFTS); *T. subulata* hydroethanolic leaf extract (HELTS). ^a^ The most intense fragment ions are described.

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
