# Peer review of "Chemical Characterization of Flowers and Leaf Extracts Obtained from Turnera subulata and Their Immunomodulatory Effect on LPS-Activated RAW 264.7 Macrophages"

_molecules, 2022, doi:10.3390/molecules27031084_

Round 1

Reviewer 1 Report

The manuscript entitled “Chemical Characterization of Flowers and Leaf Extracts Obtained from Turnera subulata and Their Immuno-modulatory Effect on LPS-Activated RAW 264.7 Macrophages” regards chemical characterization and antinflammatory effects of T. subulate flower hydroalcoholic extracts in macrophages.

Although the paper has relevant information about the beneficial effect of extract, the analytical procedures of the phytochemical should be completed. For this reason, I think that additional information should be included to improve the results displayed.

Line 100: What was the particle size obtained after grinding?

Line 101: The authors have to specify the volume of solvent used to perform the extractions procedures or at least indicate the ratio solvent/sample applied.

Line 104: The decoction may induce the degradation of thermosensitive compounds and on the other side, the extraction time is considerably higher in hydroalcoholic extractions. Why the author developed different extraction conditions for aqueous and hydroalcoholic extracts?  

Did the author evaluate the extraction yield of each extraction procedure?

 Line 110-line 112:  It is not clear how the extracts were prepared, as in the previous section they were filtered, evaporated and freeze-dried. Do the authors centrifuge the reconstituted extracts? Which extracts are treated in this way, the hydroalcoholic or the aqueous ones? Please explain this aspect better.

Line 125: blank?

Line 127: Specify the composition of calibrant.

Figure 1: What do the authors mean by 2x, 5x and 10x? It would be advisable to indicate this in the caption of the figure.

Table 1: There are some missing information. If authors developed UPLC-MS/MS, why don't they specify the molecular formula and ion fragments for each compound in the table?. Moreover, in figure 1 it could be seen that there are some isomers of vixetin and other compounds. Why do they not specify the number of isomers, if applicable?

Furthermore, with this lack of information, the identification of Vixetin 2-O-rhamnoside would be unclear, as the fragments of each isomer cannot be observed.

Since they did not use internal standards in each of the analyses, quantification of the phytochemicals is necessary to objectively evaluate the results.

In my opinion, to increase the interest of the readers and to complete the information shown in this work, I consider it necessary to incorporate the quantification of the extracts in order to know the concentration of the components in each extract.

Line 423: It is difficult to be sure that the anti-inflammatory effect of the extracts is mainly due to vixetin, as no tests have been carried out with this compound in isolation. For this reason, it would be interesting to qualify this statement or to develop trials with the isolated compound.

Author Response

Please check the attached file with all answers and justifications for the Editor and Reviewer 1.

Reviewer 2 Report

In general, this is a very well written manuscript and the topic is certainly of interest to the readership of Molecules. In this report, the authors explored the phytochemical profile of the aqueous and hydroalcoholic extracts of Turnera subulate flowers and leaves. In addition, they investigated the anti-inflammatory properties of the T. subulata extracts in vitro and test their roles in modulating the level of pro-inflammatory (TNF-α, IL-1β, and IL-6) and anti-inflammatory (IL-10) cytokines secretion as well as PGE-2 and NO production.

While this manuscript provided interesting insights in understanding the composition of T. subulata extracts, it is still not clear which is the key component contributing to the observed activity. The authors mentioned that it may be associated with vitexin, as papers have been published indicating its anti-inflammatory role in modulating level of TNF-α, IL-6, IL-1β, IL-10, PGE-2 and NO, which are the exact same sets of targets analyzed in this report. Therefore, the novelty of this manuscript is compromised.

The authors also talked about synergism for multiple times, however, is there any experimental evidence indicating the exist of synergistic effect? Which metabolites were involved in this effect? Please provide experimental data either in the manuscript or in the supplemental data to support this statement.

In addition, the authors claimed vitexin is an important compound based on their LC-MS/MS data, however, high intensity in the MS data is not able to indicate importance/high quantity in the extract. It would be better if the author could provide more quantification result, either by ELSD or simply by weight, that could lead to a more accurate fingerprint.

For figure 1, why two/three peaks are labeled with the same compound/number, eg. in figure 1A there are two peak 6 corresponded to vitexin. Please provide more characterization data to explain the difference, they can’t be the same compound. Also, the retention time of some components, such as tryptophan in figure 1A and 1C shifted a lot, please provide some explanation in the main text.

Here I have several more detailed comments regarding to this manuscript, which are listed based on page/section/line number:

Page 3, section 2.2: please provide more details for the gradient. Also please double check the mobile phase, line 116 says acetonitrile/5% formic acid was used (which is not quite common), and line 117 says acetonitrile/0.1% formic acid. Also please specify what is a white control? Is that a blank? Please deposit the MS data, and also show the comparison between the signature spectrum from GNPS and the spectrum from this experiment in the supplementary material.

Page 4, section 2.6 and 2.7: there is an extra “t” in line 178 and 190. There should have a space between the “2ug/mL" on line 172 and 182. Please confirm the use of 40 mL of supernatant on line 187, maybe there is a typo?

Page 5: please provide some explanation about what is the cosine value.

Page 5, table 1: based on LC-MS/MS analysis, it could not determine the stereochemistry, in that sense, it is not able to claim the component is the exact molecule with the listed chiral info. Please double check the info for peak 5 in HEFTS and peak 9 in HELTS, there may have typos.

For figure 3-8, the label of LPS is not correct, should include C+. Also the statistic analysis in figure 3-8, it would be more clear to only label p value between tested condition with LPS stimulation vs C+ control.

It would also be good to draw out some key structures, either in the main manuscript or in the supplementary material.

Author Response

Please check the attached file with all answers and justifications for the Editor and Reviewer 2.

Reviewer 3 Report

This manuscript is technically well done by natural products chemistry, and the pharmacological experimental approach was well conducted to assess the immunomodulatory mechanism of action of the extracts to exert the anti-inflammatory activity; however, some minor suggestions might improve their manuscript:

  1.  In the text, there are many words with capital letters as initials which are not necessary as a literary resource for style, for instance: Hydroethanolic Flower Extract, Hydroethanolic Leaf Extract, Period Cycle Time, Pulser Frequency, Accumulation Time, curtain Ga, Ion Spray Voltage Floating, Streptomycin, Penicillin, Vitexin, etc. Authors should correct some typing mistakes (no spaces between words, etc.). A revision of grammar and style would be helpful too.
  2.  The authors must correct the improper use of the term doses, dose-dependent, dose-induced, etc. Because the term "dose" is restrictive to therapeutics, clinical pharmacology, in in vivo experiments, and when tissues are perfused in vitro and exposed to a bolus application of the drug, the absolute drug concentrations are uncertain. It becomes more appropriate to specify the amount of drug administered. Meanwhile, the authors worked with isolated tissues immersed or exposed to a known MPE extract "concentration" (the same happens with cell cultures). Please review: Neubig RR, et al. International Union of Pharmacology Committee on receptor nomenclature and drug classification. XXXVIII. Update on terms and symbols in quantitative pharmacology. Pharmacol Rev, 2003;55:597-606.
  3.  Authors should show PGE2 and NO levels exerted by the concentrations of extracts without LPS stimulation in graphs 7A-D and 8A-D, just as they did in the rest of the graphs presented. Alternatively, at least explain to the extent such a decision.
  4.  A drawing of the leading chemical structures found (although all are known secondary metabolites) may be necessary to compare future structure-activity relationships.
  5.  As the authors commented in the text, the immunomodulatory effects are a substantial cause of the anti-inflammatory activity, as the title indicates; however, why did the authors not perform a fast anti-inflammatory experiment with at least the highest dose? For example, the ear edema models in mice are sensitive and reliable tests for the anti-inflammatory activity of topically applied materials; indeed, plant extracts, compounds, oils, and ointments with potential anti-inflammatory activity can be rapidly assessed in these non-invasive tests. Such experiments implicate more work for the research group, but given the number of participants in the article, this suggestion may increase the impact of their investigation.

Author Response

Please check the attached file with all answers and justification for the Editor and Reviewer 3.

Round 2

Reviewer 1 Report

The revised manuscript entitled “Chemical Characterization of Flowers and Leaf Extracts Obtained from Turnera subulata and Their Immuno-modulatory Effect on LPS-Activated RAW 264.7 Macrophages” was profoundly improved giving relevant information that supports the analytical characterization. Moreover, all considerations given by Reviewer were conducted. However, I have some little modifications prior publication.

Line 107: µg/mL instead of µm/mL

Figure 1: Overall the figure 1 was improved thoroughly, but it could be interesting to characterize the most intense peaks. For instance, in Figure 1A the most intense peak was not characterized. I think that the authors should identified it.

The new information provided by authors help to understand better the identification of compounds since the supplementary material supports this fact.

Author Response

The response file is attached. Thanks

Reviewer 2 Report

A) Given the bioactivity is not new (the anti-inflammatory and immunomodulatory effects is known for both vitexin or Turnera species), if the authors considered the identification of vitexin in T. subulata extracts is the highlight in this report, it would be better to provide a full characterization (including NMR for stereo info) and detailed quantification, otherwise the finding is weak to some extent.

B) It is not appropriate to claim synergistic effect just based on literature, please provide some experimental data to support this statement.

C) As the entire report is based on the statement that vitexin is the key component which contributes to the anti-inflammatory and immunomodulatory effects, it is necessary to include the quantification here, not report as “will be done in the future”. Please at least provide experimental result for vitexin, and it will be ideal to provide quantification for all the reported compound to construct a more accurate fingerprint.

D) It is shocked to see so many mistakes here in the key figure. For all the peaks labeled as “na”, does that mean that there is no spectrum matched in GNPS? And how you can distinguish between ferulate and isoferulate given their spectrum is the same? It is not appropriate to conclude just based on retention time.

E) I see the authors changed the 5% formic acid to 1%, however, their mobile phase is using 0.1% formic acid, why using different acid percentage for column equilibration? In addition, why use pure organic phase for equilibration, should use the initial gradient right? Is there any washing step in the method? Please provide detailed LC method here.

H) As no stereochemistry info has been confirmed, it is not appropriate to claim the identification of any molecule with chiral centers, such as vitexin-2-O-rhamnoside, 7-O-beta-glucopyranosyl-4’-hydroxy-5-methoxyisoflavone, etc. If only based on MS data, it has to be reported as vitexin-2-O-rhamnoside or its isoforms, unless there is evidence showing the enantiomer/diastereomer have distinct spectra so that to tease out these possibilities.

Author Response

The response file is attached. Thanks

Reviewer 3 Report

The authors accomplished most of the suggestions to improve their manuscript.

Author Response

The authors accomplished most of the suggestions to improve their manuscript.

We appreciate your words. As mentioned in a previous review process, your comments and suggestions were of great value in improving our manuscript.